# From Sorting Algorithms to Scalable Kernels: Bayesian Optimization in High-Dimensional Permutation Spaces

**Zikai Xie**[1][*]**& Linjiang Chen**[1][†]
[1]State Key Laboratory of Precision and Intelligent Chemistry
University of Science and Technology of China
Hefei, Anhui 230000, China
{zikaix,linjiangchen}@ustc.edu.cn

## Abstract

Bayesian Optimization (BO) is a powerful tool for black-box optimization, but its application to high-dimensional permutation spaces is severely limited by the challenge of defining scalable representations. The current state-of-the-art BO approach for permutation spaces relies on an exhaustive $\Omega(n^2)$ pairwise comparison, inducing a dense representation that is impractical for large-scale permutations. To break this barrier, we introduce a novel framework for generating efficient permutation representations via kernel functions derived from sorting algorithms. Within this framework, the Mallows kernel can be viewed as a special instance derived from enumeration sort. Further, we introduce the **Merge Kernel**, which leverages the divide-and-conquer structure of Merge Sort to produce a compact, $\Theta(n \log n)$ to achieve the lowest possible complexity with no information loss and effectively capture permutation structure. Our central thesis is that the Merge Kernel performs competitively with the Mallows kernel in low-dimensional settings, but significantly outperforms it in both optimization performance and computational efficiency as the dimension $n$ grows. Extensive evaluations on various permutation optimization benchmarks confirm our hypothesis, demonstrating that the Merge Kernel provides a scalable and more effective solution for Bayesian optimization in high-dimensional permutation spaces, thereby unlocking the potential for tackling previously intractable problems such as large-scale feature ordering and combinatorial neural architecture search.

## 1 Introduction

As one of the most widely adopted approaches to black-box optimization, Bayesian optimization (BO) (Shahriari et al., 2015) has found broad application in machine-learning hyper-parameter tuning (Wu et al., 2019), financial portfolio optimization (Gonzalvez et al., 2019), chemical and material discovery (Luo et al., 2025), and catalyst formulation design (Xie et al., 2023). BO employs probabilistic surrogate models—most commonly Gaussian processes (GPs)—to approximate the unknown objective and uses an acquisition function to balance exploration and exploitation, thereby approaching the global optimum with the fewest possible evaluations.

While BO has thrived in continuous and categorical domains (Greenhill et al., 2020; Garrido-Merchán & Hernández-Lobato, 2020; Nguyen et al., 2020; Bartoli et al., 2025), permutation spaces remain underexplored despite their ubiquity in tasks like traveling-salesman problems, robotic planning, and sequential order-of-addition experiments (Guidi et al., 2020; Lin & Rios, 2025). The core challenge lies in kernel design, while existing approaches fall into two categories:

- **General discrete BO (e.g., COMBO)**: Rely on graph encodings but struggle with the specific structural constraints of permutations (Oh et al., 2019).

---

[*]Corresponding author.
[†]Co-corresponding author.

- **Permutation-specific kernels (e.g., BOPS)**: The state-of-the-art (SOTA) Mallows kernel (Deshwal et al., 2022) uses an exhaustive $O(n^2)$ pairwise representation. While effective in low dimensions, this quadratic complexity creates massive statistical redundancy compared to all possible representations ($2^{n^2} \gg n!$) and becomes computationally intractable for large $n$.

To address the limitations of existing approaches, we propose a sorting-algorithm–driven kernel-design framework for permutations and instantiate it with the Merge kernel, which reduces the feature dimension to $O(n \log n)$—the information-theoretic lower bound for encoding a permutation. The central insight is that any comparison-based sorting algorithm is defined by a fixed sequence of element comparisons; recording the binary outcome of each comparison yields a feature vector for the permutation. Choosing an algorithm with a deterministic comparison tree—such as Merge Sort or Bitonic Sort—thus produces a representation that is both fixed in length and highly compact.

**Contributions**. Our work makes three principal contributions:

- **General framework.** We propose a unified design framework that constructs permutation-space kernels by treating any comparison-based sorting algorithm as a feature generator. Within this view, the classic Mallows kernel is recovered as the special case obtained when the framework is instantiated with enumeration sort.
- **Merge kernel.** Applying the framework to Merge Sort yields Merge Kernel, whose $O(n \log n)$ construction matches the information-theoretic lower bound on comparison complexity.
- **Comprehensive evaluation.** We assess the effectiveness of our kernels on diverse synthetic and real-world benchmarks. Results on low-dimensional benchmarks show competitive performance against the SOTA Mallows kernel, while it significantly outperforms the Mallows kernel and BO on high-dimensional benchmarks.

Our results demonstrate that the Merge kernel provides a practical and efficient tool for permutation optimization, significantly enhancing BO's applicability to diverse AI scenarios.

## 2 BACKGROUND AND RELATED WORKS

### 2.1 PERMUTATION OPTIMIZATION

Here we describe the problem formulation of permutation optimization with a fixed length $n \in \mathbb{N}$. Let $[n] = \{1, 2, ..., n\}$, a permutation is a function $\pi : [n] \longrightarrow [n]$ such that $\pi$ is bijective. The set of all permutations of $[n]$ is the symmetric group

$$\mathcal{S}_n = \big\{ \pi \mid \pi : [n] \longrightarrow [n] \text{ is bijective} \big\}.$$

We are given a costly-to-evaluate, possibly noisy black-box function $f : \mathcal{S}_n \longrightarrow \mathbb{R}$, which assigns a real-valued quality (e.g., cost, loss, reward) to every permutation $\pi$. Hence, the optimization problem can be formulated as

$$\pi^* = \arg \min_{\pi \in \mathcal{F}} f(\pi)$$

where $\mathcal{F} \subseteq \mathcal{S}_n$ is the feasible set. In this study we only consider the unconstrained case, therefore we have $\mathcal{F} = \mathcal{S}_n$; in practice $\mathcal{F}$ may exclude permutations violating domain rules.

### 2.2 BAYESIAN OPTIMIZATION

Bayesian optimization (BO) (Shahriari et al., 2015) is an optimization algorithm for black-box objective functions that no closed-form expression or gradient information is available and whose evaluation is often an expensive physical or computational experiment. The algorithm first fits the observed data with a surrogate model, most commonly a Gaussian process (GP) (Williams & Rasmussen, 2006), and then employs an acquisition function to select the next query point, balancing exploration of uncertain regions against exploitation of promising areas identified by the surrogate.

The kernel $K(\mathbf{x}, \mathbf{x}')$ is the central design lever in a Gaussian-process surrogate: it defines the similarity metric between inputs, thereby specifying the prior smoothness assumptions and, through

GP inference, the posterior mean and uncertainty. Extending BO to any new search domain is therefore tantamount to endowing that domain with an appropriate kernel function. Whereas Euclidean spaces typically rely on Gaussian (RBF) kernels, discrete structures—and permutations in particular—require bespoke constructions that faithfully encode ordering relationships. For example, (Oh et al., 2022) uses the Position kernel (Zaefferer et al., 2014) and L-ensemble with Acquisition Weights to extend BO on permutation space to a batched BO scheme.

While significant progress has been made in scaling BO to high-dimensional continuous domains (e.g., Eriksson et al. (2019); Wang et al. (2016)), extending BO to large-scale structured discrete domains like permutations presents a distinct set of challenges centered on kernel design, which is the primary focus of this work.

## 2.3 Mallows Kernel for Permutation Space

BOPS-H (Deshwal et al., 2022) is the current SOTA BO algorithm for permutation optimization, which proposes to employ Mallows kernel (Jiao & Vert, 2015) on the symmetric group $\mathcal{S}_n$ in a similar manner to the RBF kernel on the Euclidean space. The Mallows kernel $K_{Mal}(\pi, \pi')$ for the permutation pair $(\pi, \pi')$ is defined as the exponential negative of the number of discordant pairs $n_d(\pi, \pi')$ between $\pi$ and $\pi'$:

$$K_{Mal}(\pi, \pi') = \exp(-ld(\pi, \pi')) \tag{1}$$

where $l \geq 0$ is the length-scale parameter of the Mallows kernel, and $d(\pi, \pi')$ is the Kendall-$\tau$ distance (Kendall, 1938) which counts the number of pairs of elements ordered oppositely by $\pi$ and $\pi'$:

$$d(\pi, \pi') = \sum_{i<j} \left[ 1_{\pi(i)>\pi(j)} 1_{\pi'(i)<\pi'(j)} \right. \left. + 1_{\pi(i)<\pi(j)} 1_{\pi'(i)>\pi'(j)} \right] \tag{2}$$

Intuitively, the Kendall-$\tau$ distance counts the differences of all pair-wise comparisons between $\pi$ and $\pi'$. For example, let $\pi = (1, 2, 3, 4)$ and $\pi' = (2, 1, 4, 3)$. Two pairs are discordant among the six unordered pairs: $(1, 2), (3, 4)$, hence $d(\pi, \pi') = 2$ and $K_{Mal}(\pi, \pi') = \exp(-2l)$.

## 3 Merge Kernel: Generating Kernels from sorting algorithms

In the previous section, we have shown that the core of the Mallows kernel is the pairwise comparison of all elements. Equivalently, it maps a permutation $\pi$ to a feature vector

$$\Phi_{Mal}(\pi) \in \{0, 1\}^{\binom{n}{2}}$$

where each coordinate corresponds to the comparison of a pair of elements: 0 if they are in ascending order, and 1 otherwise. We then have

$$\begin{aligned} K_{Mal}(\pi, \pi') &= \exp\left(-\frac{\|\Phi_{Mal}(\pi) - \Phi_{Mal}(\pi')\|^2}{2\ell^2}\right) \\ &= K_{\mathrm{RBF}}\big(\Phi_{Mal}(\pi), \Phi_{Mal}(\pi')\big). \end{aligned} \tag{3}$$

under the reparameterization $l = \frac{1}{2\ell^2}$ for the length-scale parameter $\ell$ in the Gaussian RBF kernel. Since the RBF kernel $K_{\mathrm{RBF}}$ is strictly positive definite on $\mathbb{R}^d$ and thus satisfies Mercer's condition (Mercer, 1909), and because positive definiteness is preserved under composition with any deterministic mapping $\Phi$, it follows that $K(\pi, \pi')$ constructed from $\Phi$ also satisfies Mercer's condition and is therefore a valid kernel function.

Thus, other pairwise comparison methods can also be used to construct analogous feature vectors and yield valid kernel functions when combined with an RBF kernel. Naturally, we can extend the idea of pairwise comparison to sorting algorithms: the essence of a sorting algorithm is to compare

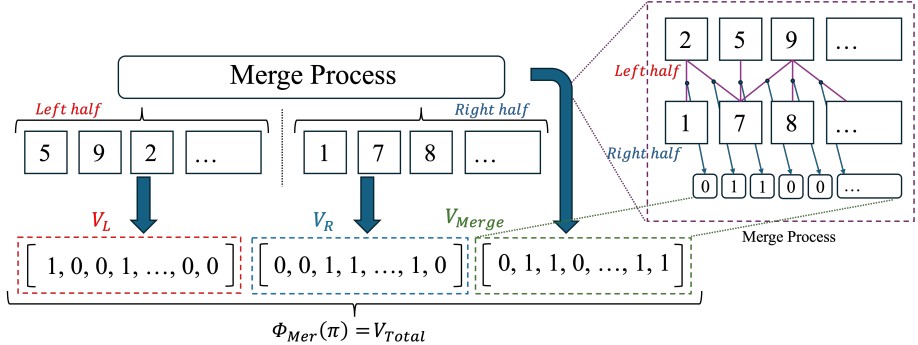

Figure 1: Diagram of generating Merge feature vector $\Phi_{Mer}$.

elements in a sequence and swap them when necessary. Consequently, each sorting algorithm embodies a unique pairwise comparison strategy, suggesting that we can build permutation-space kernels based on sorting procedures. As a sorting algorithm traverses all elements, it records whether each comparison leads to a swap, thereby fully reconstructing the original permutation; hence, the resulting feature vector retains all information without any loss. Viewed in this light, the Mallows kernel's exhaustive enumeration of every element pair can be interpreted as an enumeration-sort–inspired featurization, where enumeration sort ranks each item by comparing it with every other element and then places it directly in its final position. Therefore, we can replace the above Enumerate Sort with a sorting algorithm to generate another feature vector for permutation kernel: it is used to obtain the information on whether a swap occurred during every comparison in the comparison map.

**Constraint: Fixed Comparison Map.** To serve as a valid kernel feature vector $\Phi(\pi)$, the sorting algorithm must execute a fixed sequence of comparisons regardless of the input. Otherwise, elements at the same position in two feature vectors would represent information with different meanings, rendering comparison impossible. This excludes adaptive algorithms like Quicksort. We implement a stabilized Merge Sort that enforces redundant comparisons (totaling $L + R - 1$ per merge step). Specifically, each comparison evaluates the leading unmerged elements of the subsequences; stabilizing the path ensures that the feature at each index consistently encodes the same local structural decision, enabling meaningful distance computation.

**Merge Kernel.** We introduce the Merge Kernel, leveraging the divide-and-conquer structure of Merge Sort. The construction of the feature vector mirrors the recursive sorting process: (1) divide the sequence into two halves; (2) recursively generate feature vectors $\Phi_{Left}$ and $\Phi_{Right}$ for each half; and (3) merge the sorted halves, recording the binary outcome of each comparison into $\Phi_{Merge}$ (e.g., 0 if the element from the left half is larger, and 1 otherwise). The final feature vector $\Phi$ is formed by concatenating $[\Phi_{Left}, \Phi_{Right}, \Phi_{Merge}]$. This results in a compact representation of length $O(n \log n)$, matching the information-theoretic lower bound for lossless encoding. The kernel is defined as Equation 4 and an example of generating Merge feature is illustrated in Figure 1. The element-pair comparison mapping $\Phi_{Mer}(\pi)$ is shown in Algorithm 1. Here we present an example below for permutation [1,4,3,2] to show the **Merging** process. Detailed discussion of sorting algorithm choices is included in Appendix B.1.

$$
\begin{aligned}
K_{Mer}(\pi, \pi') &= \exp\left(-\frac{\|\Phi_{Mer}(\pi) - \Phi_{Mer}(\pi')\|^2}{2\ell^2}\right) \\
&= K_{\text{RBF}}\left(\Phi_{Mer}(\pi),\ \Phi_{Mer}(\pi')\right).
\end{aligned}
\tag{4}
$$

---

**Example: Feature Mapping for** $\pi = (1, 4, 3, 2)$

1. **Initial Split:** $\pi$ splits into $L_0 = (1, 4)$ and $R_0 = (3, 2)$.
2. **Recurse on $L_0 = (1, 4)$:** Merging sorted lists '[1]' and '[4]' yields feature vector $V_L = [0]$.
3. **Recurse on $R_0 = (3, 2)$:** Merging sorted lists '[2]' and '[3]' yields feature vector $V_R = [1]$.

---

4. **Final Merge:** Merge Sorted lists '(1,4)' and '(2,3)'. The fixed comparison path generates the merge vector $V_{Merge} = [0(1 < 2), 1(4 > 2), 1(4 > 3), 1(\text{left padding})]$.
5. **Concatenate:** The final feature vector is $\Phi_{Mer}(\pi) = V_L \oplus V_R \oplus V_{Merge} = [0] \oplus [1] \oplus [0, 1, 1, 1] = [0, 1, 0, 1, 1, 1]$.

---

**Algorithm 1** MERGE FEATURE MAPPING $\Phi_{Mer}(\pi)$

---

**Input**: Permutation $\pi$ of length $n$
**Output**: Feature vector $\Phi_{Mer}(\pi)$

    **if** length($\pi$) == 1 **then**
        **return** []
    **end if**
    **if** length($\pi$) == 2 **then**
        **return** [1] **if** $\pi[0] > \pi[1]$ **else** [0]
    **end if**
    Let $mid = \lfloor \frac{n}{2} \rfloor$
    Let $V_{Left}, V_{Right} = \Phi_{Mer}(\pi[: mid]), \Phi_{Mer}(\pi[mid :])$
    Let $\hat{\pi}_l, \hat{\pi}_r = \text{sorted}(\pi[: mid]), \text{sorted}(\pi[mid :])$
    Let $V_{Merge} = [], i = j = 0$
    **while** $i < \text{length}(\hat{\pi}_l)$ and $j < \text{length}(\hat{\pi}_r)$ **do**
        **if** $\hat{\pi}_l[i] > \hat{\pi}_r[j]$ **then**
            $V_{Merge}$.append(1)
            $j+ = 1$
        **end if**
        **if** $\hat{\pi}_l[i] < \hat{\pi}_r[j]$ **then**
            $V_{Merge}$.append(0)
            $i+ = 1$
        **end if**
    **end while**
    **return** $V_{Left} + V_{Right} + V_{Merge}$

---

We have established that Merge sort, with a specially designed fixed-comparison procedure, is uniquely capable of constructing valid kernel functions among common sorting algorithms with a complexity of $\Omega(n \log n)$. We now demonstrate that the feature vector derived from Merge Sort achieves the theoretical lower bound on vector length for lossless permutation encoding. First, note that the lower bound on time complexity for any comparison-based sorting algorithm is $\Omega(n \log n)$; as this complexity directly corresponds to the number of element comparisons during sorting, it similarly sets a lower bound on the length of the feature vector. On the other hand, consider the permutation space consisting of all $n!$ possible permutations of length $n$. From an information-theoretic viewpoint, encoding all $n!$ permutations without loss using a binary feature vector composed solely of $0, 1$ requires a minimum vector length of $\log_2(n!)$. Applying Stirling's approximation (Knuth, 1999), we have: $\log_2(n!) = n \log_2 n - n log_2 e + O(\log n) = \Omega(n \log n)$. Consequently, the feature vector length of the merge-sort-based kernel (Merge Kernel) reaches this theoretical lower bound for lossless permutation encoding.

It is worth noting that, when constructing the Merge kernel via Merge Sort, we have not required the feature mapping $\Phi$ to possess any group invariance property, such as right-invariance. Traditionally, a permutation-distance measure should be invariant under right multiplication, meaning that applying an identical right-multiplication operation to two permutations should not alter the distance between them. However, only sorting algorithms with complexity $O(n^2)$ can yield fully right-invariant kernel functions with no information loss (otherwise, a simple Spearman's footrule (Diaconis & Graham, 1977) with $O(n)$ complexity can hold right-invariance as well), since such invariance necessitates exhaustive pairwise comparisons among all $\frac{n(n-1)}{2}$ pairs of elements—an impossibility for more efficient sorting algorithms like Merge Sort. Consequently, although the Merge kernel achieves better computational efficiency through a more compact encoding, it sacrifices a certain degree of performance due to the loss of right-invariance.

We can view the relationship between the Merge and Mallows kernels as a feature selection process. Given that the $\Phi_{\text{Mer}}(\pi)$ vector corresponds to a structured subset of the complete $\Phi_{\text{Mal}}(\pi)$ feature space, incrementally adding the missing comparison positions to $\Phi_{\text{Mer}}$ is equivalent to a gradual transformation towards the $\Phi_{\text{Mal}}$ vector. This transformation represents a principled trade-off: the process of "buying back" the property of right-invariance through the incorporation of more features is achieved at the explicit cost of sacrificing computational efficiency. However, because this requires the development of appropriate analytical tools to quantify the exact marginal gain in invariance per added comparison, this remains a fascinating, yet highly complex, direction for future research that is beyond the scope of this paper.

## 4 EXPERIMENTS

### 4.1 BENCHMARKS AND EXPERIMENT SETTINGS

Our empirical evaluation focuses on the SOTA BOPS-H algorithm (Deshwal et al., 2022) as the primary control baseline. This choice is twofold: first, BOPS-H was shown to substantially outperform other permutation-specific methods like COMBO (Oh et al., 2019). Second, our core objective is a principled comparison between our Merge Kernel and the Mallows Kernel—both natively designed for permutations. This comparison serves as a direct evaluation of representation power without confounding factors from domain adaptation. Furthermore, we adapt TuRBO (Eriksson et al., 2019) as a high-dimensional BO algorithm of general purpose to evaluate the overall competitiveness of our framework. Since TuRBO is designed for continuous space, we apply a continuous relaxation to the permutation space: we define the search space as a $d-$dimensional unit hypercube $[0, 1]^d$, where the discrete permutation is induced by the argsort of the continuous vector elements.

To disentangle whether the performance of the Merge kernel stems merely from its compact dimensionality or from the specific structured information it captures, we introduce a randomized baseline. Specifically, we randomly subsample a fixed number of pairwise comparisons from the full Mallows feature vector, ensuring the total dimensionality exactly matches that of the Merge kernel. We then apply the same RBF kernel to these features. If the success of the Merge kernel were driven solely by "compression" rather than the "informative structure" of the features, this baseline should achieve comparable performance. In addition, discussion of using Spearman's footrule as featurization method is also added to Appendix B.3, due to limited space in the main manuscript. The resources and source code of BOPS algorithms are available at: https://github.com/XieZikai/MergeBO.

#### 4.1.1 LOW-DIMENSIONAL BENCHMARKS

The study by Deshwal et al. (2022) exclusively considers problems with dimensions of 30 or less. Accordingly, we adopt their experimental settings to form our suite of low-dimensional benchmarks. The BOPS-H algorithm follows and modifies the local-search strategy used in COMBO, examining only the set of neighbouring permutations of the current incumbent to restrict the combinatorial search space, we therefore adopt the same procedure in our experiments. GPyTorch (Gardner et al., 2018) and BoTorch (Balandat et al., 2020) libraries are used to implement both algorithms. Expected Improvement acquisition function is used for all the experiments, and 10 restarts are used for local search based EI optimization for BOPS-H and MergeBO. Each benchmark is evaluated with 20 independent trials, each consisting of 50 iterations. The random seed for each trial is set to its trial index.

We evaluate our method on the same two synthetic benchmarks and the same two real-world applications following Deshwal et al. (2022). Detailed information for all benchmarks are listed below:

**(1)Quadratic Assignment (QAP)**$_{n=15}$**.** This is a classic facility-location problem: assign $n$ facilities to $n$ locations so that flow costs and distances align optimally. We use the 15-city instances from **QAPLIB** (Burkard et al., 1997), each with cost matrix $A$ and distance matrix $B$, and minimise $\text{Tr}(APBP^\mathsf{T})$ over permutation matrices $P$.

**(2)Travelling Salesman (TSP)**$_{n=15}$**.** The TSP seeks the shortest Hamiltonian cycle through a set of cities and is a standard benchmark for route-planning. Our instances are 15 city PCB drill tours from **TSPLIB** (Reinelt, 1995); the score is the total travel time to visit all holes exactly once and return.

**(3)Floor Planning (FP)**$_{n=30}$**.** Floor planning is an NP-hard VLSI layout task that packs rectangular modules on a chip while minimising area and manufacturing cost. We evaluate two 15-block variants (FP-1 and FP-2); each permutation defines a block placement whose cost we minimise.

**(4)Cell Placement (CP)**$_{n=30}$**.** Cell placement arranges logic cells on a row to reduce wire-length and hence circuit delay. We consider 30 equal-height cells with a fixed net-list; the objective is the total Manhattan wire-length induced by a permutation of cell positions.

Because the publicly available implementation of the Mallows kernel (https://github.com/aryandeshwal/BOPS) does not provide the interface required to run the Rodinia's heterogeneous many-core benchmark (Che et al., 2009), we did not perform experiments on this benchmark. We note a numerical discrepancy between our replicated results and those reported in the original paper, which we attribute to subtle implementation details not specified in the publication, such as problem instance choices. However, we emphasize that within our experimental framework, both the Merge Kernel and the Mallows Kernel were evaluated under identical conditions, ensuring a fair and controlled comparison of their relative performance.

### 4.1.2 HIGH-DIMENSIONAL BENCHMARKS: TRAVELING THIEF PROBLEMS

We introduce traveling thief problems (TTP) (Bonyadi et al., 2013) as high-dimensional benchmarks, which is a defined as a combination of TSP and knapsack problem: a thief must determine a tour through $n$ cities with distance matrix $D = \{d_{ij}\}$ while simultaneously selecting items of varying weights $w_k$ and values $p_k$ to maximize profit without exceeding a knapsack capacity $W$. This structure defines a complex, hybrid search space, combining an $n$-dimensional permutation space for the city tour with a $\{0,1\}^m$ discrete space for item selection. Despite its typical application in evaluating white-box or heuristic algorithms (Polyakovskiy et al., 2014; Gupta et al., 2015), we adapt the TTP as a true black-box benchmark, providing no structural information to the optimizer.

Following the implementation in Polyakovskiy et al. (2014), we create three distinct instances based on a $n = 280$-city problem (a 280-dimensional permutation space). These instances feature a large number of items with varying properties: (1) $m = 279$ items with uncorrelated weights; (2) $m = 837$ items with bounded strong correlation in the weights; and (3) $m = 837$ items with uncorrelated weights. These benchmarks provide a strenuous test for our proposed MergeBO and the baseline BOPS-H. Notice that the TTP is a hybrid space problem, we modified both MergeBO and BOPS-H by multiplying an RBF kernel on the $\{0,1\}^m$ discrete space:

$$K_{TTP}((\pi,\sigma),(\pi',\sigma')) = K(\pi,\pi')K_{RBF}(\sigma,\sigma') \tag{5}$$

Where $\sigma$ is the item picking strategy. The neighbouring permutation search method for BOPS-H is computationally infeasible on such a vast space. Therefore, we adopt a continuous relaxation approach, treating the entire set of optimization variables $(\pi,\sigma)$ as a continuous vector for gradient-based optimization, and the result is subsequently projected back to the nearest permutation and binary vectors.

Crucially, this relaxation approach relies on the existence of a feasible projection from the continuous feature space back to the permutation space. While our Merge kernel features preserve the structural logic of the sorting algorithm to allow for valid reconstruction, the randomized baseline lacks this structural consistency (e.g., potentially inducing cyclic or conflicting comparisons). Consequently, a valid projection for the randomized baseline is ill-defined, rendering it inapplicable to this high-dimensional optimization setting.

All experiments use the EI acquisition function. Each benchmark is evaluated with 50 independent trials, each consisting of 55 iterations (5 iterations of random initialization and 50 iterations of optimization). The random seed for each trial is set to its trial index.

### 4.1.3 EVALUATION METRICS

In this study, we employ two evaluation metrics: the final simple regret and the area under the best-so-far regret curve (AUC). In optimization, regret is defined as the difference between the best objective value observed to date and the global optimum:

$$r_t = f_t^{\text{best}} - f^*$$

Table 1: Feature length comparison of Merge and Mallows kernel over problems of different scales.

| Problem | Dimension | Merge feature length | Mallows feature length |
|---------|-----------|----------------------|------------------------|
| TSP | 15 | 45 | 105 |
| QAP | 15 | 45 | 105 |
| FP | 30 | 119 | 435 |
| CR | 30 | 119 | 435 |
| TTP | 280 | 2009 | 39060 |

Following this concept, the final simple regret is the regret value obtained in the last iteration and reflects the algorithm's ultimate optimization capability when computational cost is disregarded. By contrast, the regret AUC, or cumulative regret—the sum (area under the curve) of the simple regret across all iterations—quantifies the convergence speed of the entire optimization process:

$$\text{AUC}_T = \Sigma r_t$$

Generally speaking, the final simple regret and AUC represent different, valuable aspects of an algorithm's performance: the former represents the the quality of the solution it ultimately finds while the latter represents the "journey", or convergence speed during optimization. For sample-efficient methods like BO, these two evaluation metrics are standard practices as they allow for a comprehensive comparison.

Table 2: Performance comparison between MergeBO, BOPS-H (Mallows kernel), BOPS-H with random comparisons and TuRBO. Underlined results indicate the best numerical results in terms of mean value ± standard deviation of all trials, and bold font indicates statistically significant superiority of MergeBO against BOPS-H as determined by a binomial sign test (p < 0.05, corresponding to more than 15 wins of 20 trials).

| Problem | Simple final Regret | | | | Regret Wins | | |
|---------|------|---------|--------|-------|-------------|------|--------------|
| | Merge | Mallows | Random | TuRBO | Merge Wins | Ties | Mallows Wins |
| $\text{TSP}_{n=15}$ | $0.077 \pm 0.125$ | $\underline{0.013 \pm 0.039}$ | $0.329 \pm 0.332$ | $1.213 \pm 0.879$ | 1 | 12 | 7 |
| $\text{QAP}_{n=15}$ | $14.9 \pm 5.5 \times 10^3$ | $\underline{8.1 \pm 4.1 \times 10^3}$ | $18.1 \pm 3.6 \times 10^3$ | $14.2 \pm 5.9 \times 10^3$ | 1 | 3 | **16** |
| $\text{FP}_{n=30}$ | $24.0 \pm 9.7$ | $30.1 \pm 12.8$ | $35.7 \pm 11.2$ | $\underline{20.5 \pm 8.4}$ | 10 | 4 | 6 |
| $\text{CR}_{n=30}$ | $\underline{6.1 \pm 2.2}$ | $6.1 \pm 3.0$ | $52.15 \pm 19.0$ | $33.85 \pm 15.7$ | 9 | 3 | 8 |
| $\text{TTP1}_{n=280}$ | $\underline{23.0 \pm 11.3 \times 10^3}$ | $88.9 \pm 7.5 \times 10^3$ | | $54.8 \pm 12.6 \times 10^3$ | **50** | 0 | 0 |
| $\text{TTP2}_{n=280}$ | $\underline{14.9 \pm 7.2 \times 10^4}$ | $56.5 \pm 6.1 \times 10^4$ | | $36.8 \pm 9.4 \times 10^4$ | **50** | 0 | 0 |
| $\text{TTP3}_{n=280}$ | $\underline{8.0 \pm 3.2 \times 10^4}$ | $28.1 \pm 2.8 \times 10^4$ | | $19.1 \pm 3.6 \times 10^4$ | **50** | 0 | 0 |

| Problem | Best so far AUC | | | | AUC Wins | | |
|---------|------|---------|--------|-------|-----------|------|--------------|
| | Merge | Mallows | Random | TuRBO | Merge Wins | Ties | Mallows Wins |
| $\text{TSP}_{n=15}$ | $527.6 \pm 162.8$ | $\underline{428.2 \pm 121.9}$ | $559.7 \pm 224.9$ | $877.2 \pm 352.4$ | 5 | 0 | **15** |
| $\text{QAP}_{n=15}$ | $38.3 \pm 8.4 \times 10^5$ | $\underline{27.5 \pm 7.4 \times 10^5}$ | $42.5 \pm 6.1 \times 10^5$ | $46.7 \pm 9.2 \times 10^5$ | 1 | 2 | **17** |
| $\text{FP}_{n=30}$ | $8097.5 \pm 2163.7$ | $8665.7 \pm 2638.8$ | $9481.0 \pm 2086.2$ | $\underline{5932.1 \pm 1636.9}$ | 10 | 1 | 9 |
| $\text{CR}_{n=30}$ | $5495.6 \pm 687.7$ | $\underline{5350.5 \pm 910.1}$ | $13970.8 \pm 2408.8$ | $10340.5 \pm 2477.3$ | 8 | 0 | 12 |
| $\text{TTP1}_{n=280}$ | $\underline{20.5 \pm 4.4 \times 10^5}$ | $48.5 \pm 3.1 \times 10^5$ | | $40.0 \pm 4.9 \times 10^5$ | **50** | 0 | 0 |
| $\text{TTP2}_{n=280}$ | $\underline{12.3 \pm 3.2 \times 10^6}$ | $30.5 \pm 2.4 \times 10^6$ | | $25.9 \pm 3.8 \times 10^6$ | **50** | 0 | 0 |
| $\text{TTP3}_{n=280}$ | $\underline{6.7 \pm 1.4 \times 10^6}$ | $15.2 \pm 1.3 \times 10^6$ | | $13.2 \pm 1.5 \times 10^6$ | **50** | 0 | 0 |

It is important to note that this study does not include a comparison of wall-clock computation times. This is a deliberate choice grounded in the fundamental premise of BO, where the cost of function evaluations (e.g., physical experiments or complex simulations) is assumed to far outweigh the computational cost of the algorithm itself. Consequently, our analysis prioritizes metrics related to sample efficiency, which is the primary bottleneck in such real-world scenarios. As a more stable and implementation-agnostic proxy for computational complexity, we instead report the feature vector dimensions generated by each kernel in Table 1, which directly reflects the compactness of the learned representations. Furthermore, as our experiments were conducted on a shared high-performance computing (HPC) cluster, reported wall-clock times would be subject to scheduler-induced variability, making them an unreliable metric for rigorous algorithmic comparison.

Instead, we can report a rough time estimation based on local, small-scale experiments here: the Merge kernel is approximately 10% slower than the Mallows kernel in low-dimensional problems. This is because the Mallows kernel's calculation relies on two clean for-loops, whereas the Merge

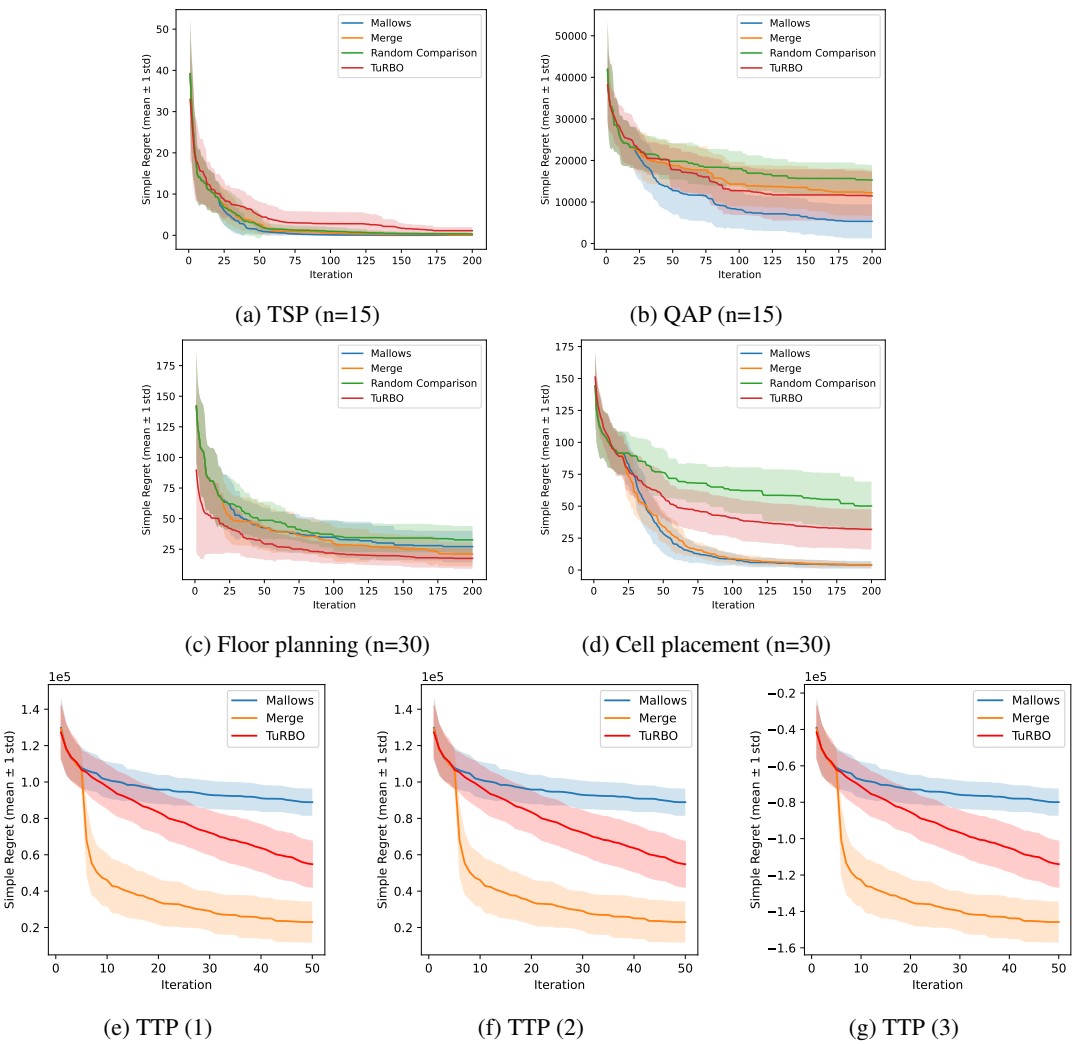

Figure 2: Low- and high-dimensional results comparing Mallows kernel (BOPS-H) and Merge kernel (MergeBO) on the current regret value (difference between best-so-far and optimal value) vs. number of iteration. Solid lines show the average regrets, while the shaded areas denote one standard deviation.

kernel requires recursive calls to Merge Sort, which involves significant constant overhead from function calls, Python list slicing, and deepcopy operations. However, in high-dimensional problems, we believe the kernel value computation bottleneck arising from the $O(n^2)$ feature space will lead to significant performance degradation.

## 4.2 RESULTS AND DISCUSSION

Our experimental results, as presented in Table 2 and Figure 2, systematically reveal a strong correlation between the performance advantage of the Merge kernel and the problem's dimensionality. For low-dimensional permutation problems (TSP and QAP), the Mallows kernel, which is specifically designed for such tasks, exhibited a slight performance advantage. However, on problems of intermediate dimensionality (FP and CR), the performance of the two kernels was comparable, with no statistically significant difference observed. This trend shifted decisively on the higher-dimensional TTP problems, where the Merge kernel demonstrated definitive superiority. The statistical data in Table 2 indicates that the Merge kernel significantly outperformed the Mallows kernel across all TTP instances on both the final regret and AUC metrics. Furthermore, the convergence curves in

Figure 2 confirm its substantially faster convergence speed. This validates the superior performance and scalability of our proposed method on complex, high-dimensional optimization problems.

The consistent underperformance of the randomized baseline confirms that unstructured feature selection fails to capture necessary permutation similarities. This validates that our method's efficiency stems from preserving principled structural information, rather than mere dimensionality reduction. Regarding TuRBO, while it demonstrates scalability by outperforming BOPS-H in high-dimensional tasks, it remains significantly inferior to our method. This substantial gap underscores the necessity of our specialized permutation optimization framework over generic continuous relaxation strategies and general high-dimensional optimization approaches.

These results corroborate our core hypothesis: the Merge kernel possesses an inherent advantage in high-dimensional permutation optimization problems, owing to its more compact structural design. The disparity between its $O(n \log n)$ feature complexity and the Mallows kernel's $O(n^2)$ complexity widens dramatically as the dimensionality n increases. This is explicitly quantified by the feature length comparison in Table 1: for low, intermediate, and high-dimensional problems, the feature dimensionality of the Mallows kernel is approximately 2, 3.5, and 19.5 times that of the Merge kernel, respectively. These results indicate that the experimental performance is governed by a trade-off between two key factors: (1) the ability to capture global information via the distance metric, and (2) the search efficiency driven by the compactness of the feature space. Due to its right-invariance property, the Mallows kernel possesses a stronger distance metric capability than the Merge kernel. However, as dimensionality increases, the vast disparity in feature vector length leads to a more pronounced space-compression effect. The resulting gains in search efficiency begin to outweigh the performance benefits afforded by the superior distance metric. Consequently, as the problem dimensionality continues to grow, the performance of the Merge kernel ultimately surpasses that of the Mallows kernel by a significant margin. This naturally suggests a dimension-dependent heuristic for practitioners: leveraging the Mallows kernel's robust, right-invariant distance metric for low-dimensional tasks, while switching to the Merge kernel to capitalize on its superior scalability in high-dimensional regimes.

## 5 CONCLUSIONS

In this work, we proposed a novel kernel construction framework for permutation spaces by leveraging sorting algorithms as structured comparison schemes. Within this framework, we introduced the Merge kernel—an efficient, compact, and theoretically grounded alternative to the quadratic Mallows kernel. We showed that the Merge kernel achieves the information-theoretic lower bound on feature complexity while preserving meaningful structural information. This contribution bridges sorting theory and kernel design, revealing a fundamental trade-off between a model's structural invariance and the compactness of its feature space.

Empirical experiments confirmed this trade-off: while the BOPS-H algorithm held a marginal advantage in low-dimensional problems, their performances were comparable on intermediate-dimensional tasks. In high-dimensional settings, however, MergeBO's compact representation enabled far superior search efficiency, allowing it to significantly outperform BOPS-H and TuRBO. This work opens exciting possibilities for scaling Bayesian optimization to larger permutation spaces. Future directions include exploring other sorting algorithms with constant comparison counts or constructing stabilized sorting algorithm of $O(n \log n)$ complexity for kernel design, and applying this framework to challenging real-world domains like combinatorial neural architecture search and computational biology.

## 6 ACKNOWLEDGEMENTS

The computational resources for this work were provided by the robotic AI-Scientist platform of the Chinese Academy of Sciences (CAS). The authors thank Dr. Xenophon Evangelopoulos (University of Liverpool) for insightful early discussions and for raising questions regarding right-invariance, which helped clarify the limitations of earlier approaches.

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

## A  THE USE OF LARGE LANGUAGE MODELS

We declare that the large language models (LLMs) ChatGPT and Gemini are only used to aid and polish writing. No further applications of LLMs are used in this research, including but not limited to retrieval, research ideation and experiment designing.

# B  EXTRA DISCUSSIONS

## B.1  DISCUSSION ON SORTING ALGORITHM CHOICES

Thanks to the fixed comparison map constraint, not every sorting algorithm can induce a valid feature mapping suitable for kernel construction. This is because the mapping from permutations to feature space must yield feature vectors of fixed length; otherwise, feature vectors of differing lengths would not be compatible with the RBF kernel. Hence, only sorting algorithms that have a fixed comparison path and a constant number of comparisons across all inputs can generate valid feature mappings, e.g., a fixed sorting network with a predetermined comparator sequence Batcher (1968). This strict constraint allows us to exclude the vast majority of $O(n \log n)$ complexity sorting algorithms that are stochastic or adaptive, such as quicksort, heapsort, and standard Merge Sort, since their comparisons are data-dependent, not fixed. Nevertheless, Merge Sort could be an exception: although typically Merge Sort ceases comparisons once all elements from one subsequence have been merged, redundant comparisons can be artificially introduced during the merge procedure, resulting in a fixed number of comparisons $L + R - 1$, where $L$ and $R$ are the length of the two subsequences. This ensures that both the comparison path and the number of comparisons remain identical across different permutations, thus establishing a fixed comparison path.

Stabilizing the comparison map of other $O(n \log n)$ sorting algorithms is quite challenging, and other $O(n^2)$ sorting algorithms are equivalent to Mallows kernel. At the moment we can confirm that Bitonic Sort satisfy the above constraint: the comparison maps of permutations are fixed and identical according to their length. Given its $O(n \log^2 n)$ complexity, we excluded this algorithm from the present study, but we consider it a viable direction for future extensions of this framework.

## B.2  DISCUSSION ON INFORMATION-THEORETIC LOWER BOUND

It is important to clarify that the feature vector lower bound discussed in this paper refers to the information-theoretic lower bound on the number of pairwise comparisons required to reconstruct the relative order of two permutations, i.e., the lower bound implied by lossless information compression. This should be distinguished from the algorithmic lower bound for computing a distance regarding to the original permutations. For example, the Mallows kernel relies on the Kendall–$\tau$ distance, and although the latter can be computed in $O(n \log n)$ time using algorithms such as modified Merge Sort, this does not reduce the information requirement to $O(n \log n)$. Such algorithms still implicitly depend on the relative order of all $O(n^2)$ pairs of elements, but accelerate computation by batch processing rather than by reducing the underlying information model. In contrast, our method does not require explicit access to all $O(n^2)$ pairs. Instead, it achieves a complete reconstruction of the relative order using only $O(n \log n)$ pairwise comparisons.

## B.3  DISCUSSION ON SPEARMAN'S FOOTRULE

To evaluate the trade-off between permutation information, dimension reduction and right-invariance, we employ Spearman's footruleDiaconis & Graham (1977) distance as another benchmark baseline. Similar to Euclidean measures on raw coordinates, Spearman's footrule operates directly on the permutation group $S_n$. Formally, for two permutations (rankings) $\sigma$ and $\pi$ of $n$ elements, the distance is defined as the sum of the absolute differences between the ranks of each element:

$$d_{\text{footrule}}(\boldsymbol{x}, \boldsymbol{x}') = \sum_{i=1}^{n} |r_i - r_i'| \tag{6}$$

Obviously, the corresponding featurization method $\Phi_{footrule}(\pi)$ is an identical mapping that directly uses the permutation $\pi$ as its feature vector. The computational complexity of Spearman's footrule is $O(n)$, and it is fully right-invariant since it simply calculates the $L_1$ distance between two vectors.

However, while this mapping retains the raw rank values, it treats permutations merely as vectors in a Euclidean space, thereby ignoring the underlying algebraic structure of the symmetric group.

Table 3: Performance comparison between MergeBO, BOPS-H (Mallows kernel), BOPS-H with random comparisons, TuRBO and Spearman's footrule. Underlined results indicate the best numerical results in terms of mean value $\pm$ standard deviation of all trials.

| Problem | Simple final Regret | | | | |
|---|---|---|---|---|---|
| | Merge | Mallows | Random | TuRBO | Spearman |
| $\text{TSP}_{n=15}$ | $0.077 \pm 0.125$ | $\underline{0.013 \pm 0.039}$ | $0.329 \pm 0.332$ | $1.213 \pm 0.879$ | $0.026 \pm 0.052$ |
| $\text{QAP}_{n=15}$ | $14.9 \pm 5.5 \times 10^3$ | $\underline{8.1 \pm 4.1 \times 10^3}$ | $18.1 \pm 3.6 \times 10^3$ | $14.2 \pm 5.9 \times 10^3$ | $10.2 \pm 4.7 \times 10^3$ |
| $\text{FP}_{n=30}$ | $24.0 \pm 9.7$ | $30.1 \pm 12.8$ | $35.7 \pm 11.2$ | $\underline{20.5 \pm 8.4}$ | $34.6 \pm 9.0$ |
| $\text{CR}_{n=30}$ | $6.1 \pm 2.2$ | $6.1 \pm 3.0$ | $52.15 \pm 19.0$ | $33.85 \pm 15.7$ | $\underline{1.5 \pm 2.4}$ |
| $\text{TTP1}_{n=280}$ | $\underline{23.0 \pm 11.3 \times 10^3}$ | $88.9 \pm 7.5 \times 10^3$ | | $54.8 \pm 12.6 \times 10^3$ | $88.7 \pm 10.1 \times 10^3$ |
| $\text{TTP2}_{n=280}$ | $\underline{14.9 \pm 7.2 \times 10^4}$ | $56.5 \pm 6.1 \times 10^4$ | | $36.8 \pm 9.4 \times 10^4$ | $56.2 \pm 6.2 \times 10^4$ |
| $\text{TTP3}_{n=280}$ | $\underline{8.0 \pm 3.2 \times 10^4}$ | $28.1 \pm 2.8 \times 10^4$ | | $19.1 \pm 3.6 \times 10^4$ | $28.4 \pm 2.5 \times 10^4$ |

| Problem | Best so far AUC | | | | |
|---|---|---|---|---|---|
| | Merge | Mallows | Random | TuRBO | Spearman |
| $\text{TSP}_{n=15}$ | $527.6 \pm 162.8$ | $428.2 \pm 121.9$ | $559.7 \pm 224.9$ | $877.2 \pm 352.4$ | $\underline{397.1 \pm 98.0}$ |
| $\text{QAP}_{n=15}$ | $38.3 \pm 8.4 \times 10^5$ | $\underline{27.5 \pm 7.4 \times 10^5}$ | $42.5 \pm 6.1 \times 10^5$ | $46.7 \pm 9.2 \times 10^5$ | $31.8 \pm 8.5 \times 10^5$ |
| $\text{FP}_{n=30}$ | $8097.5 \pm 2163.7$ | $8665.7 \pm 2638.8$ | $9481.0 \pm 2086.2$ | $\underline{5932.1 \pm 1636.9}$ | $9187.7 \pm 2024.1$ |
| $\text{CR}_{n=30}$ | $5495.6 \pm 687.7$ | $5350.5 \pm 910.1$ | $13970.8 \pm 2408.8$ | $10340.5 \pm 2477.3$ | $\underline{4673.4 \pm 1020.15}$ |
| $\text{TTP1}_{n=280}$ | $\underline{20.5 \pm 4.4 \times 10^5}$ | $48.5 \pm 3.1 \times 10^5$ | | $40.0 \pm 4.9 \times 10^5$ | $48.2 \pm 4.3 \times 10^5$ |
| $\text{TTP2}_{n=280}$ | $\underline{12.3 \pm 3.2 \times 10^6}$ | $30.5 \pm 2.4 \times 10^6$ | | $25.9 \pm 3.8 \times 10^6$ | $30.3 \pm 2.4 \times 10^6$ |
| $\text{TTP3}_{n=280}$ | $\underline{6.7 \pm 1.4 \times 10^6}$ | $15.2 \pm 1.3 \times 10^6$ | | $13.2 \pm 1.5 \times 10^6$ | $15.3 \pm 9.4 \times 10^6$ |

Unlike the Mallows kernel or our proposed Merge kernel, which embed specific probabilistic or hierarchical priors, this naive representation doesn't project the permutation to a compact manifold and fails to capture the compact dependencies within the feasible space. Nevertheless, its $L_1$ nature allows it to effectively approximate local structural discrepancies through simple summation.

The experiment results of Spearman's footrule are added in Table 3 above. In low-dimensional settings, Spearman's footrule performs slightly worse than the Mallows kernel but marginally better than the Merge kernel. This can be attributed to the high relevance of right-invariance in lower dimensions, where accurately measuring the similarity between local regions—which often share similar performance characteristics—is critical. Notably, the method exhibits rapid convergence on TSP and CR tasks, stemming from the inherent local additivity of these problems. However, Spearman's footrule offers lower discriminative resolution for permutations compared to the Mallows kernel. This coarser granularity tends to bias the search towards exploitation rather than exploration; while this enables quick convergence to local optima, it may limit the model's ability to escape them for a global solution.

Conversely, in high-dimensional problems, Spearman's footrule performs comparably to the Mallows kernel but significantly lags behind the Merge kernel. This stark contrast highlights the Merge kernel's superior capability to compress high-dimensional search spaces and accelerate optimization through hierarchical decomposition.

