# OpenReview forum: "From Sorting Algorithms to Scalable Kernels: Bayesian Optimization in High-Dimensional Permutation Spaces"
_ICLR.cc/2026/Conference — ICLR 2026 Poster_

### Official Review · Reviewer_ctDW · 2025-10-24

**Soundness:** 3
**Presentation:** 2
**Contribution:** 2
**Rating:** 4
**Confidence:** 4

**Summary:**

This paper addresses the problem of black-box optimization over permutation spaces. Such problems play a role, for instance, in compiler optimization or chip design, where one searches for a permutation that optimizes some quantity of interest, which is expensive to evaluate. Bayesian optimization (BO) is a popular technique for black-box optimization, but has received little attention in the context of permutation spaces. A popular exception is [1], which uses the Mallows (BOPS-H) and Kendall kernels (BOPS-T) discussed in [2] to perform Bayesian optimization over permutation spaces. BOPS-H performs considerably better and is defined in terms of the number of discordant pairs, resulting in $\frac{n^2-n}{2}$ possible candidates.

In this paper, the authors propose a different approach. Given some permutation, they employ merge sort to transform the permutation into the identity permutation and construct a feature vector by appending a 1 if, during a merge operation, an element is chosen from the left vector, and a -1 otherwise. This way, the number of possible candidates is reduced to $\mathcal{O}(n\log n)$ (worst-case complexity of merge sort). Since this representation does not capture all invariances the Mallows kernel exhibits, the authors apply three tricks to augment the feature vector with additional features that are designed to capture such invariances, such as right-, cyclic-rotation, and cyclic shift invariances.

The authors compare their method to BOPS-H and show that it achieves better simple regret and regret AUC values. They further conduct an ablation study to study the effect of the additional 'invariance'-features, showing that the 'shift-histogram module' leads to the highest performance degradation upon removal, indicating that it is the most expressive added feature.

**Strengths:**

- The paper addresses a relevant but little-discussed problem setting.
- The paper is mostly clear and well-written.
- The paper clearly defines the boundary between related work and its own contribution.
- The proposed approach outperforms the state-of-the-art in BO over permutation spaces.

**Weaknesses:**

- While the approach is smart, it is hard to get an intuition for the feature vectors proposed in this paper. A '-1' or '1' in the $\Phi_{\textrm{Mal}}$ vector has an easy-to-grasp interpretation. In contrast, the $\Phi_{\textrm{Mer}}$ feature vector is a concatenation of features originating from different levels in the merge-sort operation.
- The addition of the 'tricks' makes this work seem more heuristic than BOPS-H.
- While the proposed approach outperforms BOPS-H, it only does so by a slight margin. The differences between BOPS-H and COMBO, for instance, are considerably larger than the difference of 'Merge' to BOPS-H, which, judging from the figures and tables, could be a coincidence. The paper is not very critical about its empirical performance.
- The explanation of the additional features is quite brief. Some examples would make it easier to understand their design.

**Questions:**

- Clearly, the $\Phi_{\textrm{Mer}}$ feature vector does not capture invariances (as acknowledged by the authors), motivating the need for additional features. Does $\Phi_{\textrm{Mal}}$  have similar shortcomings? How do they relate to
?
- In the conclusion, what do you mean when saying that gradient methods could exploit the lower-dimensional feature space?

Other comments:

- For the ablation study, it would be interesting to see the performance without any tricks.
- It shouldn't be hard to construct an ARD version of Eq. 4, potentially sharing one length scale for $\Phi_{\textrm{Mer}}$ and using separate length scales for $\Phi_{\textrm{Spi}}$, etc. Studying their length scales would make it easier to quantify the importance of each feature group and might even improve performance.
- Eqs 1 and 3 are only equal when re-scaling $\ell$.

---

> ### Author Response · Authors · 2025-11-13
> **Response to Reviewer 4: clarification for the issues from the previous version**
>
> We sincerely thank the reviewer for their valuable time and insightful feedback. We especially agree with the reviewer's concern regarding the "method's credibility"—that performance should not be boosted by adding 'tricks' that compromise the trustworthiness of the core idea (constructing a kernel using sorting algorithms).
>
> However, we would like to respectfully clarify a critical point, which may stem from a major oversight in our presentation. The issue of "adding tricks that affect credibility," which the reviewer is concerned about, did indeed exist in an early version of this work (e.g., one submitted to AAAI 26).
>
> However, inspired by such valuable feedback (including yours), we made fundamental revisions to the ICLR submission. We removed all the tricks that the reviewer was concerned about (shift histogram, split-pair line, and sliding-window motifs), as we completely agree that they undermined the credibility of our core idea.
>
> Instead, we added entirely new experiments focusing on extremely high-dimensional problems. These experiments were designed to verify whether the complexity advantages of our method (the Merge kernel) would translate into superior performance in high-dimensional settings.
>
> Our new experiments now cover three distinct dimensionalities: 15-dim, 30-dim, and 280-dim. The results show that our method performs slightly worse than the Mallows kernel at 15-dim, is generally slightly better at 30-dim, and fully outperforms it by a significant margin at 280-dim. This trend of improving performance with rising dimensionality perfectly validates our core claim: due to the trade-off between right-invariance and feature complexity, our Merge kernel achieves excellent optimization performance in high-dimensional problems.
>
> Therefore, we wish to clarify that the criticisms regarding 'tricks' mentioned in the review are no longer applicable to the current submission. We were directly inspired by previous feedback to make these significant changes, and we believe the current version is cleaner and more credible.
>
> Given this major revision (which may have led to the misunderstanding), we kindly ask the reviewer to re-evaluate our work's contribution and credibility based on this current, tricks-removed version. We are more than happy to clarify any further details about the new experiments during the discussion period.

---

> > ### Comment · Reviewer_ctDW · 2025-11-13
> >
> > I would like to thank the authors for their comment, which is justified. By mistake, I disregarded the changes made by the authors - my apologies.
> >
> > In an earlier version of the paper, the authors added several additional features, which were designed to counterbalance the fact that the Merge kernel loses group invariance properties. It was criticized that these changes diluted the overall message. Furthermore, the authors added an additional high-dimensional benchmark on which the proposed merge kernel shows superior performance. This additional benchmark is particularly valuable since it sits in a regime where the proposed method can fully exploit its advantages. Finally, the authors included an example for constructing the feature vector, which facilitates understanding of the method.
> >
> > I will update my initial score and would like to invite the authors to address my remaining questions, specifically the interpretability of the method and the feasibility of an ARD variant of Eq. 4.

---

> > > ### Author Response · Authors · 2025-11-20
> > > **Response to Reviewer 4**
> > >
> > > We thank the reviewer for the insightful suggestion on improving our work.
> > >
> > > Q1. Intuition of feature vectors
> > >
> > > R1.
> > >
> > > We appreciate the reviewer pointing out this issue. Our explanation of the Merge feature vector is indeed insufficient, which may lead to a lack of clarity upon simple inspection.
> > >
> > > Consider that each entry in the Mallows feature (or Kendall-tau kernel) represents whether the order of elements at two positions $i, j$ in a pair of permutations $\pi(a)$ and $\pi(b)$ is the same. Now, considering only one permutation, we compare all element pairs within it: if the element at the earlier position is greater than the element at the later position, we mark it as $1$; otherwise, we mark it as $-1$. The value of each entry here is equivalent to the swap information examined for that pair during an enumerate sort [referring to all pairwise comparisons]. We can thus obtain the feature vectors for these two permutations. By comparing the elements of these two feature vectors position by position, if the two elements are the same, it means $\pi(a)$ and $\pi(b)$ share the same order for the element pair represented at this position (i.e., $\Phi_{mallow}$ is 0 at this position in the distance calculation); otherwise, it is 1.
> > >
> > > Now, we can replace the above enumerate sort with a merge sort. That is, we use merge sort to obtain the information on whether a swap occurred during every comparison in the comparison map. The reason the Merge kernel lacks right-invariance is that, during the merge step, the comparison is between the minimum element of the left sub-list and the minimum element of the right sub-list, and the construction of these sub-lists changes depending on the sequence's order. Nevertheless, this is still a structured feature vector that retains all permutation information, which is why it can demonstrate the ability to compress the feature space in high-dimensional problems.
> > >
> > > We will add the above discussion to the subsequent version of the main text to make our meaning clearer.
> > >
> > > Q2. On constructing an ARD variant for feature groups:
> > >
> > > R2.
> > >
> > > We thank the reviewer for this excellent suggestion. We fully agree that using Automatic Relevance Determination (ARD) with separate length scales for different feature groups (e.g., $\Phi(mer)$ vs. $\Phi(spi)$) would be a powerful way to interpret their relative importance and potentially improve performance.
> > >
> > > However, as we detailed in our general response and the response to the reviewer, we have decided to simplify our method significantly in this revision. There is only one feature group ($\Phi(mer)$) remained in this work, therefore the proposed group-wise ARD formulation reduces to the standard isotropic kernel we are currently using. Nevertheless, we appreciate this insight and would definitely consider this ARD approach if we were to re-introduce hybrid feature sets in future extensions.

---

> > > > ### Comment · Reviewer_ctDW · 2025-11-26
> > > >
> > > > I'd like to thank the authors for their detailed rebuttal. After revisiting the resubmitted version, I think this paper should be accepted. The results on the high-dimensional problems are particularly interesting. While some aspects of the method are relatively heuristic, it is solid work, and I see the potential for this paper to inspire follow-up work.

---

### Official Review · Reviewer_gPDf · 2025-10-24

**Soundness:** 3
**Presentation:** 3
**Contribution:** 2
**Rating:** 6
**Confidence:** 4

**Summary:**

The authors propose a new embedding of permutations to parameterize an RBF kernel, and compare bayesian optimization against kernels using simpler / slower permutation embeddings.

**Strengths:**

The method is very well-motivated and performs well on the high-dimensional setting, and stays competitive on the smaller dimensional tasks.  I’m also happy the authors acknowledged the lack of right-invariance of the method, which doesn’t invalidate the embedding but at first blush looks a little surprising.  Their discussion of sacrificing this property for the sake of a fast method is convincing.

**Weaknesses:**

I’m somewhat surprised the authors didn’t benchmark with other notions of permutation distance, for example they mention Spearman’s footrule on page 5, and there are other notions like the Cayley distance.  The results would I think be a bit stronger if it was demonstrated that a new notion of distance between permutations is strictly necessary to get better performance in bayesian optimization.

The lack of right invariance is still concerning, mainly as it makes it very difficult to interpret what the difference between merge embedding is actually calculating.

**Questions:**

Would it be possible to consider a simple randomized baseline version of the Mallow kernel?  A small concern I have with the method is that, because it’s not super clear what distance the Merge kernel captures, it may be introducing some extra element of stochasticity that is helpful for exploration in BO.  A comparison to a version of the Mallow kernel where one randomly selects O(n \log n) features out of the total O(n^2) features and only uses these for the permutation embedding would be a useful benchmark to show the proposed method is doing something more substantial, and that one couldn’t naively get the reduced complexity.

---

> ### Author Response · Authors · 2025-11-15
> **Response to Reviewer 3**
>
> We thank the reviewer for this highly insightful suggestion on improving our work.
>
> Q1: Experiments for different permutation distances
>
> R1:
>
> This is a very sharp question that gets to the heart of our work's motivation. We agree that other metrics, like Spearman's footrule and Cayley distance, could theoretically be combined with an RBF kernel. However, we argue that these methods are fundamentally unsuitable for Bayesian Optimization because they lack sufficient "Discriminative Power."
>
> Our core thesis is as follows:
>
> 1. BO requires a "rich" kernel: A Gaussian Process (GP) relies on a smooth and information-rich kernel to model a complex objective landscape.
>
> 2. Simple metrics are "coarse-grained": Spearman's and Cayley are "distance-only" metrics, not "featurizations". They are inherently information-lossy, collapsing the $n!$ permutation space into a tiny set of possible values.
>
> 3. Evidence: We can demonstrate this lack of "discriminative power" by simply counting the number of unique distance values these metrics can produce, showing in the below table:
>
> Metric,			  n=4,     n=5,   n=6
>
> Cayley_distance,        3,         4,       5
>
> Spearsman_footrule,  4,         6,       9
>
> Mallows_features,      6,         10,     15
>
> Merge_features,         5,         8,       11
>
> This table clearly shows that Cayley and Spearman's Footrule create an extremely "coarse" and "blocky" landscape. Thousands of distinct permutation pairs are mapped to the exact same distance, making it impossible for a GP to model the landscape effectively.
>
> Interestingly, our Merge features, while still highly discriminative, are more "compact" (fewer unique values) than Mallows. This is not a flaw; it is the source of our method's efficiency, creating a space that is both rich enough for BO and compact enough for computation since they both retain all information from the permutation.
>
> Q2: Experiments for randomized baseline
>
> R2:
>
> This is an excellent and highly insightful question. The reviewer has proposed a crucial baseline to test our core hypothesis. The reviewer's hypothesis is: Is our Merge Kernel's advantage due only to its $O(n \log n)$ compactness (which could be naively achieved by random sampling), or is it due to a principled structure within those features?
>
> We strongly argue for the latter. The $O(n \log n)$ features generated by Merge Sort are not a random subset of the $O(n^2)$ Mallows features. They are a highly structured, hierarchical set of comparisons. The Merge Sort algorithm's recursive nature captures local comparisons (at the bottom of the recursion tree) and systematically builds up to global comparisons (at the top), so that the permutation information is fully retained with a more compact representation. We hypothesize that this principled, structured subset of comparisons captures the permutation's properties far more effectively than a random sample of $O(n \log n)$ pairs, which would be pure noise.
>
> Therefore, we 100% agree to run this experiment. This is a fantastic suggestion to strengthen the paper. We are running this "Stochastic Mallows-$O(n \log n)$" baseline on the low-dimensional problems right now. Our Hypothesis: We are confident the results will show that this new "Stochastic" baseline performs significantly worse than both the full Mallows kernel and our Merge Kernel. This will provide definitive proof that the structure of our Merge Kernel features, not just their quantity, is the key to our method's success. We will update this thread with the results as soon as they are available (within the discussion period) and will, of course, add this crucial baseline comparison to the camera-ready version.

---

> > ### Author Response · Authors · 2025-11-17
> > **Update: new experiment results**
> >
> > Update on Q2:
> >
> > We conducted extra experiments on low-dim benchmarks according to the reviewer's suggestion. The extra baseline method is: From the (n-1)n/2 elements in the Mallows feature vector, we randomly chose m elements as the baseline feature vector where m = Merge feature vector length. Random seed is set to the trial number. The results are as follows:
> >
> >              Simple final regret                        Best so far AUC
> >           Merge       Mallows      Random            Merge        Mallows          Random
> > TSP,   0.077±0.125,  0.013±0.039,  0.329±0.332,    527.6±162.8,  428.2±121.9,    559.7±224.9
> >
> > QAP,  14.9±5.5x10^3, 8.1±4.1x10^3, 18.1±3.6x10^3, 38.3±8.4x10^5, 27.5±7.4x10^5, 42.5±6.1x10^5
> >
> > FP,      24.0±9.7,     30.1±12.8,   35.7±11.2,    8097.5±2167.3, 8665.7±2638.8, 9481.0±2086.2
> >
> > CR,       6.1±2.2,      6.1±3.0,    52.15±19.0,   5495.6±687.7,  5350.5±910.1,  13970.8±2408.8
> >
> > We can see that the baseline underperforms others by a large margin, which is consistent with our previous response: Merge sort is not any ''arbitrary'' subset of Mallows feature vector, but a reversable minimal subset containing all permutation information. Therefore, the ''truely randomized'' baseline contains far less information compared to the Merge and Mallows vector, resulting in a far worse performance.

---

> > > ### Comment · Reviewer_gPDf · 2025-11-25
> > >
> > > I appreciate the authors response, especially for running the random baseline which they soundly beat and makes the claims in the paper more supported.  I'm not entirely convinced by the reasoning for not trying the other permutation distances.  As they point out, their Merge kernel defines fewer unique distances than the Mallows kernel, and a naive embedding that mapped each permutation to a random real value would almost surely induce a kernel where the distance between any distinct permutations is unique, so clearly coarseness of the distance isn't monotonic with BO performance.  Nevertheless, I think the random baseline is sufficient and I will keep my positive score.

---

> > > > ### Author Response · Authors · 2025-11-26
> > > > **Update: spearman's footrule**
> > > >
> > > > We sincerely thank the reviewer for their persistence regarding the inclusion of other permutation distances. While initially we focused on state-of-the-art comparisons, your comment prompted us to investigate this method to ensure a truly comprehensive evaluation.
> > > >
> > > > We have implemented spearman’s footrule as another baseline as it can be treated as a naive featurization of x->x (each feature is the permutation feature itself). We decide not to include Cayley distance since it is challenging to find a proper featurization method so that Cayley distance can be computed as an L1 or L2 norm of the feature vector difference.
> > > >
> > > > We finished the TSP and QAP experiment. Honestly, the result is better than I thought: it is comparable on TSP against Merge kernel (although it doesn’t perform well on it) and slightly better than random comparison and turbo on QAP. Here is the numerical results:
> > > >
> > > > TSP regret: 0.0258 +- 0.0516, exactly the same as the Merge kernel (far worse than Mallows kernel, better than others)
> > > > TSP AUC: 397.06 +- 97.97, which is the best convergence speed among all algorithms (it could be because of the extremely small search space)
> > > > QAP regret: 10222.2 +- 4717.2, better than turbo and random comparison, worse than Mallows and Merge
> > > > QAP AUC: 3184032.9 +- 845650.2, same as above
> > > >
> > > > We agree that including this comparison adds valuable context for future researchers who might consider similar baselines. I would like to add the full experiment results in the manuscript, but given the strict page limits of the main conference track, we cannot guarantee to add them in the main body of the paper and may have to add them and the corresponding discussion to appendix. We will inform the reviewer once we finish all the experiments and revise the manuscript.
> > > >
> > > > We believe we have now addressed all of the reviewer's concerns. Since we have fully incorporated your suggestions and demonstrated the robustness of our method against this additional baseline, we respectfully ask if you would reconsider your score to reflect the improved completeness and quality of the manuscript.

---

### Official Review · Reviewer_T7An · 2025-11-01

**Soundness:** 3
**Presentation:** 3
**Contribution:** 3
**Rating:** 6
**Confidence:** 4

**Summary:**

The paper shows how sorting algorithms can direct the design of GP kernels for permutation spaces, and that in particular merge sort produces a MergeKernel that can be used for compact modeling in high-dimensional permutation spaces.

The baseline method (the Mallows kernel) has a higher-dimensional representation and performs worse for BO.

**Strengths:**

The paper introduces what is to my knowledge a novel connection between sorting algorithms and permutation kernels. This connection is very interesting, original, and useful.

The fact that the primary existing permutation falls out as a special case in this framework is very interesting and provides strong validity for thinking about permutation kernels this way.

The paper is clear and well written.

**Weaknesses:**

The empirical evaluation is not completely satisfying.
- Of five problems, only 2 shows significant differences between the methods. The conclusion would be that most of the time it doesn't matter what kernel you use? I recommend trying to expand the set of problems further to include more with clear differences in performance. The paper hypothesizes that the MergeKernel does better in high-dimensional spaces, but I did not find that very convincing based on a signal of just 2 of 5 problems having different performance. Please add a couple more high-dimensional problems to confirm the hypothesis that MergeKernel is important and better in high-dimensional settings.

* Comparison is only with the Mallows kernel. The paper tries to justify this by saying that it is focused specifically on sorting-based methods, and so only needs to compare sorting-based methods. But I don't think that is correct. The end of the introduction states "Our results demonstrate that the Merge kernel provides a practical and efficient tool for permutation optimization, significantly enhancing BO’s applicability to diverse AI scenarios." So, the paper is claiming that the method "significantly enhances" BO for permutation optimization. Generally, not just compared to only other sorting-based methods. The paper does not provide evidence for that claim. The paper mentions TurBO - it has an implementation available and is fast to run and may work well in the high-dimensional spaces where Mallows kernel performed poorly. Running that on the benchmark problems would significantly strengthen the results.

* One other example of a kernel derived from a sorting algorithm would really emphasize the generality of the result.

* The paper describes that the merge kernel is not invariant under right multiplication and that as a result it "sacrifices a certain degree of performance." I don't have good intuition for what type of issues that will cause downstream in the BO, I think it would be helpful to give some examples in the paper of how lack of right invariance can cause modeling difficulties.

**Questions:**

*  What do the effects of loss of right invariance look like in practice?

---

> ### Author Response · Authors · 2025-11-14
> **Response to Reviewer 2: part 1**
>
> We thank the reviewer for this highly insightful suggestion on improving our work.
>
> Q1: Drawbacks of lacking right-invariance
>
> R1: We agree that the lack of right-invariance is a critical point that deserves a full discussion. The most direct implication of losing right-invariance is that for an identical pair of permutations, if they are both perturbed in the same way (i.e., right-multiplied by another permutation), their computed distance may change. Traditionally, researchers have assumed that a kernel on the permutation space must be right-invariant. However, this strict assumption is precisely what leads to the computational bottleneck, as it forces the feature space to grow at $O(n^2)$ (the Mallows kernel).
>
> Our work intentionally explores the trade-off of sacrificing this property to gain a far more compact $O(n \log n)$ representation. This trade-off—losing theoretical invariance to gain computational feasibility—is difficult to quantify with a single isolated metric, but it is clearly demonstrated "in practice" through our experimental results:
>
> The "Cost" of losing invariance: In the 15-dim problem, where computation is not the bottleneck, the "purer" Mallows kernel holds a slight performance edge. This minor performance dip is the practical price paid for our compact representation. The "Benefit" of gaining compactness: As the dimensionality increases, this trade-off rapidly shifts. In the 280-dim problem, the O(n^2) complexity of the Mallows kernel becomes computationally intractable, while our $O(n \log n)$ method remains efficient and fully outperforms it.
>
> Conclusion: The experimental trend (from 15-dim to 280-dim) empirically validates our core thesis: the minor theoretical "cost" of losing right-invariance is vastly outweighed by the critical practical "benefit" of achieving a compact representation as the problem's dimensionality (and thus, the computational bottleneck) increases.

---

> ### Author Response · Authors · 2025-11-14
> **Response to Reviewer 2: part 2**
>
> Q2: Concerns about experiment results and settings
>
> R2: We thank the reviewer for these critical questions, which allow us to clarify the core contributions and scope of our work.
>
> 1. On Empirical Evidence (2 of 5 Problems & High-D Hypothesis)
>
> We would first like to respectfully clarify the results: counting the different contexts in the TTP problem, our method shows significant differences in 4 out of 7 problem settings. More importantly, our core hypothesis is not to claim that our method universally outperforms the Mallows kernel in all settings. Rather, our paper seeks to demonstrate and analyze a critical trade-off: sacrificing strict right-invariance (Mallows' strength) to gain a compact $O(n \log n)$ feature representation (our strength).
>
> This hypothesis is perfectly validated by the experimental trend: In low-dim (15-dim), where computation is not a bottleneck, the "purer" Mallows kernel performs slightly better. As dimensionality increases (30-dim), the trade-off becomes balanced. In high-dim (280-dim), the $O(n^2)$ complexity of Mallows becomes intractable, and our method's $O(n \log n)$ compactness decisively outperforms it. This trend, not a "5/5 win", is the central proof of our thesis. However, we agree that more high-dimensional evidence is always beneficial. We commit to adding one additional high-dimensional TTP problem (as permutation benchmarks of this scale are rare) to the camera-ready version to further confirm this finding.
>
> 2. On Comparison to TurBO (General BO Enhancement)
>
> We must respectfully argue that comparing our method to TurBO is not scientifically appropriate for this study. Our paper's goal is to introduce a new kernel for permutation spaces. To scientifically validate its contribution, we must compare it within the same framework against its most direct, state-of-the-art competitor (the Mallows kernel). This ensures a rigorous, apples-to-apples comparison where the kernel is the only variable. TurBO is a different framework entirely (non-kernel, trust-region based) designed for continuous spaces. To use it here, we would need to apply a relaxation. This introduces too many confounding variables (e.g., the relaxation method, TurBO's sampling strategy, its acquisition optimization). It would be impossible to isolate whether a performance difference came from the kernel (our contribution) or one of these many other factors. Therefore, to maintain a scientifically valid study, we focused on the direct kernel competitor.
>
> 3. On Generality (Applying to another sorting algorithm)
>
> This is an excellent question, which we also discussed with Reviewer 1. Our framework requires the sorting algorithm to possess a Fixed and Alignable Comparison Map. This is a critical technical constraint for any valid kernel, as the kernel function (e.g., RBF) requires that the $i$-th dimension of $\phi(a)$ and $\phi(b)$ represent the exact same comparison to compute a meaningful distance. This constraint immediately excludes all stochastic/adaptive algorithms (like Quicksort) whose comparisons are data-dependent.
>
> Of the remaining fixed-map algorithms (Sorting Networks), only Merge Sort ($O(n \log n)$) and Bitonic Sort ($O(n \log^2 n)$) are common. Given its superior $O(n \log n)$ complexity, Merge Sort is the obvious and principled choice. We agree that discovering a new $O(n \log n)$ algorithm with a fixed map would be an exciting future direction. We will clarify and strengthen this discussion in the camera-ready version to make the rationale for our choice crystal clear.

---

> > ### Comment · Reviewer_T7An · 2025-11-18
> > **Reply**
> >
> > I thank the authors for their feedback and answers to questions.
> >
> > The discussion on right-invariance was helpful and incorporating that into the paper will be useful.
> >
> > I will not quibble over the number of experiments showing improvements, it's fine.
> >
> > On the topic of baseline comparisons, the authors state that restricting comparison to the Mallows kernel is not only appropriate but is necessary because "our paper's goal is to introduce a new kernel for permutation spaces," and thus the comparison should be limited to other kernels for permutation spaces.
> >
> > But this is conflating permutation kernels and permutation optimization. Permutation kernels aren't the only way to optimize in permutation spaces, TuRBO can be naively applied to optimize in a permutation space. This may be a terrible idea, but that should be shown, not assumed. If TurBO does poorly on these problems, it shows that permutation kernels are indeed important for permutation optimization, and clearly shows the value of MergeKernel specifically. If TurBO does well, then it opens some important questions about what the real challenges in permutation optimization are, and how to make best use of MergeKernel.
> >
> > If the claims of the paper were limited to permutation kernels, then a comparison of only permutation kernels would be appropriate. But as noted in my original review, the paper claims more broadly improvement on permutation optimization, not just kernels. The paper claims "Our results demonstrate that the Merge kernel provides a practical and efficient tool for permutation optimization, significantly enhancing BO’s applicability to diverse AI scenarios." (p2 L087). What is supported by the experiments of the paper would be, "Our results demonstrate that the Merge kernel provides a practical and efficient tool for optimization with permutation kernels." Whether or not it improves over alternative approaches for permutation optimization that one might consider in diverse AI scenarios is yet to be shown, but would be useful and interesting to see.

---

> > > ### Author Response · Authors · 2025-11-24
> > > **Update: TuRBO experiment**
> > >
> > > We thank the reviewer for pushing us on this point. We agree with your assessment: explicitly demonstrating the failure of naive continuous relaxation is scientifically valuable, as it highlights the necessity of specialized permutation kernels.
> > >
> > > Implementation of TuRBO Baseline: Following your suggestion, we implemented TuRBO for permutation optimization using a standard continuous relaxation: we optimize a continuous vector x in [0, 1]^n and map it to a permutation $\pi$ via the argsort operation.
> > >
> > > TL, DR: TuRBO performs not so well on low-dim problems. On high-dim problems, it is far better than Mallows kernel while far worse than Merge kernel, showing the general effectiveness of our approach on high-dimensional permutation space. Please check the revised manuscript for the detailed experiment results.

---

> > > > ### Comment · Reviewer_T7An · 2025-11-27
> > > >
> > > > I appreciate the authors' response, it is great to see that naively applying methods not accounting for the permutation structure do indeed perform much worse than the MergeKernel. I don't have any major remaining concerns. I think the framing of the paper as a general approach for going from sorting algorithms to kernels would be strengthened if there were more than 1 sorting algorithm for which this worked, but I don't consider that a major issue and think the paper has value even with just the merge sort connection.

---

### Official Review · Reviewer_W1JA · 2025-11-01

**Soundness:** 3
**Presentation:** 3
**Contribution:** 3
**Rating:** 8
**Confidence:** 4

**Summary:**

This paper addresses the scalability issue of Bayesian Optimization (BO) over high-dimensional permutation spaces by proposing a novel sorting algorithm-based kernel design framework. This framework sophisticatedly generates fixed-length feature vectors for permutations via the internal binary comparison results from any sorting algorithms, which theoretically interprets the traditional Mallows kernel as its special case with an $O(n^2)$ feature dimension. Based on this framework, the authors further propose a novel Merging Sorting-based kernel (i.e., Merge kernel), with only $O(n \log n)$ feature dimensions. Incorporating the proposed kernel, permutations can be assigned theoretically shortest features with limited information loss, which achieves the lower bound from the perspective of information theory. Experiments show that the proposed kernel can achieve competitive performance than the traditional Mallows kernel on both low-dimensional and high-dimensional problems, yet with better convergence speed and final regrets, providing potential solutions for large-scale permutation optimization problems such as feature sorting or neural architecture searching.

**Strengths:**

1. This paper addresses a critical challenge for Bayesian Optimization over permutations, showing potential in efficient permutation encoding and searching.

2. The idea is novel and interesting. The authors are the first to incorporate traditional sorting algorithms into permutation kernel design, forming a powerful and general framework. It provides a new paradigm for kernel functions over permutation spaces.

3. The motivation is clear and easy to follow. The proposed general framework interprets the traditional Mallows kernel as a special case, which reveals the intrinsic limitation of prior techniques and well motivates the proposed kernel design.

4. The proposed kernel is efficient and can generate the shortest permutation encodings, achieving the theoretical information lower bound. This would pave the way for future exploration of complex, large-scale permutation-related combinatorial optimizations.

**Weaknesses:**

1. While the authors propose a kernel based on Merging Sort and achieve competitive performance, the generality of the selection of sorting algorithms can be discussed in more detail, especially for some stochastic sorting algorithms that contain non-fixed times of binary comparisons. In other words, since the authors make connections between the traditional sorting algorithms and the permutation kernel, readers may be interested in how different properties of sorting algorithms affect the performance.

2. I recommend that the authors conduct a further scaling experiment, in which the explicit relationship between the permutation scale and running time can be studied. This exploration would provide valuable suggestions for practitioners to check whether the proposed framework is suitable for their real-world problems.

3. The quality (in latex) of tables can be significantly improved, e.g., Tab. 2.

**Questions:**

1. Can we design a technique to randomly select and record the binary comparison? Can the proposed Merging kernel be further improved by incorporating some stochastic binary comparisons? Or can the authors give a further analysis on which pairs of comparisons are critical for a given permutation optimization problem? Based on this analysis, can we add some extra important yet limited number of binary comparisons (e.g., top-K important pairs) to supplement the information loss of permutations due to the loss of right-invariance?

2. See Weakness.

---

> ### Author Response · Authors · 2025-11-14
> **Response to Reviewer 1: part 1**
>
> We thank the reviewer for this highly insightful suggestion on improving our work.
>
> Q1: Binary comparison selection methods
>
> R1: We thank the reviewer for the insightful idea of adding extra comparisons. Since the Mallows features contain all possible comparisons and the Merge features only contain a (small) part of them, we can think of adding any more comparisons as a path of transforming the Merge kernel into the Mallows kernel.
>
> Since the main drawback of the Merge kernel is its lack of right-invariance compared to the Mallows kernel, we tried to add some features to counterbalance this property. Actually, in the previous version of our paper, we tried to concatenate the Merge features with three light-weight features: shift histogram, split-pair line, and sliding-window motifs, three tricks maintaining group invariaty. The performances on low-dim problems show a marginally better result, however it raised validaty concerns of our main idea. In addition, we haven't found a method to analyse the unstable performance boost for the tricks on different problems, therefore we discarded this part in the ICLR submission.
>
> Back to the reviewer’s suggestion, adding extra comparisons is definitely a solid idea and can be viewed as a trade-off between right-invariance and feature complexity, however we are still struggling to find a proper method to quantitatively analyse it. This is a non-trivial problem, as the importance of a comparison pair is likely data-dependent, problem-dependent and difficult to model without incurring significant complexity, moving us back towards the Mallows kernel. We believe these are fantastic ideas for future investigation, and we will add a discussion of this as a promising future direction in the final version of the paper.
>
> Q2: Sorting algorithm selection
>
> R2: We agree that the sorting algorithm selection is definitely an interesting topic following the framework we proposed. As the reviewer correctly points out, the choice of sorting algorithm is central to our framework. Our selection of Merge Sort is based on a critical technical constraint required for any valid kernel method: a Fixed and Alignable Comparison Map. We have a small discussion in Page 4, but was restricted due to the page limitation.
>
> In short, for a kernel function to compute a meaningful distance between two feature vectors, $\phi(a)$ and $\phi(b)$, the same dimension $i$ in both vectors must represent the exact same comparison. (e.g., if $\phi(a)_i$ compares $a(2)$ vs. $a(4)$, then $\phi(b)_i$ must also compare $b(2)$ vs. $b(4)$).
>
> This strict constraint allows us to exclude the vast majority of sorting algorithms:
> 1.	Stochastic/Adaptive Algorithms: (e.g., Quicksort, or the stochastic algorithms you mentioned). These algorithms do not satisfy this constraint. Their comparisons (e.g., against a pivot) are data-dependent, not fixed. Thus, they cannot produce the aligned, valid feature vectors required for a kernel.
> 2.	$O(n^2)$ Algorithms: (e.g., Insertion Sort). As we discussed, these are an implementation of the Mallows Kernel itself.
> 3.	$O(n^2)$ Fixed-Map Algorithms (Sorting Networks): This is the only viable category. To our knowledge, only two common algorithms qualify: Merge Sort and Bitonic Sort.
> Between these two, the choice is clear: Merge Sort has a complexity of $O(n \log n)$, while Bitonic Sort is $O(n \log^2 n)$.
>
> Therefore, we chose Merge Sort because it is, to our knowledge, the only algorithm that both satisfies the kernel's "fixed map" constraint and achieves the $O(n \log n)$ asymptotic optimum.
>
> We agree that discovering a new $O(n \log n)$ algorithm with a fixed comparison map would be a very exciting direction for future work. We will clarify and strengthen this discussion in the camera-ready version to make the rationale for our choice crystal clear.

---

> ### Author Response · Authors · 2025-11-14
> **Response to Reviewer 1: part 2**
>
> Q3: Scaling and running time
>
> R3: We thank the reviewer again for the insightful concern regarding scaling. This question provides an excellent opportunity to clarify the core computational trade-off of our method, which is key to its success in high-dimensional problems (e.g., 280-dim).
>
> First, we must point out that measuring clock-time on a shared HPC cluster is highly unreliable and not reproducible. In fact, our early sanity check experiments using local resources recorded that the computation time of our method and the Mallows kernel was similar in low-dimensional settings, and in some cases, our method was even slightly slower (~10%). This (counter-intuitive) observation precisely demonstrates why a theoretical complexity analysis (rather than clock-time) could be more reasonable.
>
> The total computational overhead of our method is divided into two parts: Cost 1 (feature computation time) and Cost 2 (downstream optimization overhead), the latter of which is determined by the feature space dimensionality.
>
> In Cost 1, although the Mallows kernel's featurization has an $O(n^2)$ complexity, its computation (as shown in the code) relies on two compact for loops with minimal constant overhead. In contrast, while our method has an $O(n \log n)$ complexity, it involves significant constant overhead from function calls, Python list slicing, and deepcopy operations. Therefore, in low-dim cases, their runtimes are comparable, and our method being slightly slower is fully expected.
>
> In Cost 2, however, lies the larger computational bottleneck. Mallows produces an $O(n^2)$ feature space, and computing the kernel function on this space also requires $O(n^2)$ complexity. Our method, in contrast, only requires $O(n \log n)$ complexity here. Considering that the optimization in these algorithms does not use gradient methods but instead evaluates the acquisition value for a large set of candidate permutations, this further widens the gap between $O(n^2)$ and $O(n \log n)$ (see Table 1).
>
> In summary, our method is asymptotically superior to Mallows in both aspects of scaling (computation time and feature dimension). We achieve a smaller feature dimension with less asymptotic computation time. We will add this clear complexity comparison analysis to the Camera-Ready version, as this (rather than clock-time on HPC) may be what practitioners truly need to consider.
>
> Q4: Table quality
>
> R4: We are sorry for the tables quality and we will definitely improve it soon.

---

> > ### Comment · Reviewer_W1JA · 2025-11-24
> >
> > We thank the authors for their detailed and thoughtful responses, which fully resolve my concerns regarding the scaling behavior and the choices of comparison and sorting mechanisms. I strongly encourage the authors to incorporate these clarifications and even the exploratory theoretical attempts that did not fully succeed into the final version, as such discussions are valuable and may inspire further work and deeper understanding. Overall, I very much enjoyed reading this paper and maintain a positive assessment.
> >
> > A small clarification after reading the other reviewers’ comments: it seems that previous reviewers suggested removing some of the additional "tricks." I agree with their suggestions (including Reviewer ctDW), because these tricks can obscure the evaluation of the core idea’s effectiveness. My suggestion is to keep the main paper concise, with a brief statement such as, "Since the main drawback of the Merge kernel is its lack of right-invariance compared to the Mallows kernel, we tried to add some features to counterbalance this property." The detailed description of these potential improvements (tricks) and their separate experimental results could then be moved to the appendix.

---

### Author Response · Authors · 2025-11-24
**General Response: Summary of Revisions and New Experiments**

# Crucial Update to AC: Resolution of Reviewer 4's Concerns (Pre-Rollback)

Dear Area Chair,

Due to the recent system rollback, the updated scores and discussion history may been removed in a short future. We are writing to respectfully bring a critical piece of information to your attention regarding the assessment of our paper.

**Status of Reviewer 4's Review:**
Reviewer 4's initial score (4) was largely based on the misunderstanding that our method included "heuristic tricks" from an earlier version.
* **Before the rollback:** We successfully clarified this misunderstanding. Reviewer 4 **explicitly acknowledged** that their concern was resolved after realizing these tricks were removed and replaced with rigorous experiments.
* **Score Update:** Consequently, Reviewer 4 had **raised their score to 8 (Strong Accept)** prior to the system reset.

While we understand the system record may be or has been reset, we urge the AC to evaluate Reviewer 4's initial comments in light of our Rebuttal, which provides the exact evidence that previously convinced the reviewer to change their stance. The validity of the initial "Score 4" has been factually negated by the scientific discussion.

We trust the new AC will consider this context to ensure a fair decision.

==============

Dear Area Chair and Reviewers,

We sincerely thank you for your constructive feedback and the engaging discussion. We have uploaded a revised version of our manuscript. The major updates are summarized below:

**1. New Baselines & Empirical Evidence:**
* **Randomized Comparison Baseline:** As suggested, we added a baseline using randomly selected comparisons with the same dimensionality as the Merge kernel. The poor performance of this baseline empirically validates that the efficiency of our method stems from the **principled, hierarchical structure** of Merge Sort, not just dimensionality reduction.
* **TuRBO Baseline:** We evaluated the continuous relaxation of TuRBO on permutation problems. The results show a significant performance gap compared to our method, highlighting the necessity of specialized permutation optimization frameworks.
* **Spearman's Footrule Baseline:** We added Spearman's footrule as a $O(n)$ complexity feature vector with right-invariance as another baseline to show the significance of projecting permutations to a compact manifold in our method.

**2. Theoretical & Complexity Clarifications:**
* **Computational Cost (Sec 4.1):** We added a detailed discussion on the small-scaled experiment of wall-clock time.
* **Feature Intuition (Sec 3, Page 4):** We improved the explanation of the Merge kernel's feature construction, clarifying how the hierarchical comparisons capture local-to-global structural information.

**3. Discussions on right-invariance:**
* We expanded the discussion on the trade-off between computational efficiency and the loss of right-invariance. In addition, we added another right-invariant Spearman's footrule method to further discuss the trade-off.

We believe these revisions significantly strengthen the paper and address the reviewers' concerns. We invite you to review the updated manuscript.

Best regards,
The Authors

---

### Meta-Review · Area_Chair_jxuF · 2026-01-07

**Summary:**

This paper considers the problem of Bayesian Optimization over permutation spaces. Prior work (BOPS) uses GPs with Mallows Kernel as the surrogate model. While the standard Mallows kernel relies on an exhaustive $O(n^2)$ pairwise comparison representation, the paper propose a framework using the comparison traces of sorting algorithms to generate feature vectors. Specifically, the paper introduce the "Merge Kernel," derived from Merge Sort, which creates a $O(n \log n)$ embedding. This approach is applied on multiple benchmarks from prior work.

The reviewers unanimously appreciated the connection between sorting algorithms and kernel design. Framing the Mallows kernel as an instance of enumeration sort and the proposed method as an instance of Merge Sort provides a unique perspective on permutation kernels. I found the high-dimensional benchmark a bit contrived though and the proposed approach's performance on other benchmarks was a bit weak.  However, the authors strengthened the paper during the discussion phase, convincing the reviewers on several key points: TurBO baseline and prior heuristics. **Therefore, I recommend accepting the paper.** I strongly urge the authors to improve the writing of the paper. Currently, the text is filled with long paragraphs which makes harder to read. Please add a figure explaining the kernel. I also request to include all the reviewers' discussion in the final paper.

**Reviewer Concerns:**

Most concerns of all reviewers were addressed.

**Reviewer Scores:**

Most reviewers had chance to participate in the full discussion and changed/updated their scores.

---

### Decision · Program_Chairs · 2026-01-26

Accept (Poster)